# Criterion Validity of Catapult ClearSky T6 Local Positioning System for Measuring Inter-Unit Distance

**DOI:** 10.3390/s20133693

**Published:** 2020-07-01

**Authors:** Ryan W. Hodder, Kevin A. Ball, Fabio R. Serpiello

**Affiliations:** Institute for Health and Sport (IHES), Victoria University, Footscray VIC 3011, Australia; Kevin.Ball@vu.edu.au (K.A.B.); fabio.Serpiello@vu.edu.au (F.R.S.)

**Keywords:** player tracking, team sports, tactical, validity, positioning, performance analysis, local positioning system

## Abstract

The validity of a local positioning system (LPS) to measure inter-unit distance was investigated during a team sport movement circuit. Eight recreationally active, female indoor team-sport players completed a circuit, comprising seven types of movements (walk, jog, jump, sprint, 45° change of direction and shuffle), on an indoor court. Participants wore a receiver tag (ClearSky T6, Catapult Sports) and seven reflective markers, to allow for a comparison with the reference system (©Vicon Motion Systems, Oxford Metrics, UK). Inter-unit distance was collected for each combination of participants. Validity was assessed via root mean square error, mean bias and percentage of variance accounted for, both as an overall dataset and split into distance bands. The results presented a mean root mean square error of 0.20 ± 0.05 m, and mean bias detected an overestimation for all distance bands. The LPS shows acceptable accuracy for measuring inter-unit distance, opening up opportunities to utilise player tracking for tactical variables indoors.

## 1. Introduction

### 1.1. Player Tracking Technology

Electronic performance and tracking systems (EPTS) have recently seen large developments, allowing them to facilitate tracking of players both indoors and outdoors with greater accuracy. Until recently, EPTS were limited to outdoor sports, whereby the Global Positioning System (GPS) was used to track a player’s locomotion, position and speed [1,2]. With a large focus on GPS in outdoor sports, the accuracy of this type of technology has been extensively investigated, showing adequate accuracy for assessing players’ distance [1]. However during unstructured movements, high rates of change in velocity and at peak speeds, less favourable results were found. Overestimations of up to 19.3% were found in changes in velocity during decelerations [3], while peak speeds were underestimated compared to the reference system by 14–29% [4]. Even with the considerations of GPS accounted for, use of GPS has been inaccessible for indoor sports, with stadium infrastructure preventing clear signals to satellites. As such, indoor sports including basketball, handball, futsal and netball have been restricted to expensive optical tracking systems, which are susceptible to player occlusion errors [5]. However, recent developments in local positioning systems (LPS), specifically bluetooth-based and ultra-wideband (UWB) technologies, have opened up opportunities to integrate EPTS in indoor sports [6,7,8].

### 1.2. Ultra-Wideband

Local positioning systems operate indoors usually on short-range communication between radio wave generators and receivers. As such, they can run on varying bandwidths depending on the technology, and UWB specifically occupy a large frequency bandwidth (>500 MHz) [9]. Operating at this bandwidth allows UWB systems penetration through objects such as wood, plastic, brick, and other building materials, excluding metal [10]. This provides the ability for the tracking of humans without occlusion errors occurring [9]. Additionally, unlike other radio frequency-tracking technologies, UWB’s high bandwidth combined with very low short pulses waveforms, allows for reduced signal interference from other radio frequency devices and increased signal stability [10,11].

Local positioning systems using UWB technology have been validated previously for analysing position, speed and distance [8,12,13]. During linear drills, mean and peak acceleration and speed displayed errors in the range of 0.5% to 7.5% [12], while total distance was found to have error ranges between 0.5% to 2.3% [8,12]. Finally, position was found to have the least amount of error, with 0.19% to 0.58% error ranges [13]. However, this paper used known distance as a reference, which may underestimate the difference between systems. Conversely, research pertaining to the reliability of UWB systems for indoor sports is currently limited, however one study has assessed the inter-unit reliability of one system (WIMU PRO™, RealTrack Systems, Almeria, Spain) for position, reporting an intra-class correlation coefficient for x-coordinate and y-coordinate of 0.65 and 0.88 respectively [14]. Additionally, reliability measures of the system were not significantly affected during replication of typical movements of team sport which reached speeds >15 km/h [9]. With validation of UWB for locomotion and player position, research has now focused on describing the match and training demands of each sport [15,16,17,18].

However, recent research using GPS positional data has focused on spatiotemporal tactical variables to analyse team collective behaviours and dynamics. Now with the same ability as GPS to track player positions on an indoor court, positional data from UWB systems can be used to provide contextual information to analyse players’ tactical roles and how they impact other players’ performance [19].

### 1.3. Tactical Variables

Tactical variables are used to explain player, team and opposition dynamics on the field through their interactions, spacing, synchronisation and integration alongside technical and physical variables. Tactical variables can be understood as variables that occupy both space and time (i.e., spatiotemporal) and are derived from the field of geometry [20]. Thus, spatiotemporal data, when used to describe collective behaviours of players, can provide a measure of tactical performance [21]. The most basic tactical variable is inter-player distance; identifying the distance between players’ positions on the field [22,23] and providing insight into their interactions and coordination tendencies. Team-based tactical variables include surface area and dominant region; explaining the effective playing space a team or group of players look at controlling [24,25], stretch index and length per width ratio; indicating the contraction and expansion of a team as they move through the transitional phases of a game [26,27,28]. Finally, the collective behaviour and synchronisation of a team has previously been analysed using the team centroid and approximate entropy; describing the behaviour and centre position of a team of players and their inter-player coordination respectively [25,29].

These tactical variables have previously been analysed for outfield sports using GPS [28,30]. While research is beginning to utilise LPS and optical EPTS, for tactical analysis indoors [6,7,26], these systems have yet to be validated for their accuracy in measuring inter-unit distance. This is of importance for tactical analysis, as most tactical variables are primarily made up of inter-unit distances which are then combined to create team level variables. If inter-unit distance accuracy is poor, this may compound when calculating multiple inter-unit distances for larger spatiotemporal variables. As such, the aim of this study is to assess the criterion validity of the Catapult ClearSky T6 local positioning system for measuring inter-unit distance, applicable to all indoor sports for tactical analysis.

## 2. Materials and Methods

### 2.1. Participants

Eight recreationally active female indoor team sport players (26.9 ± 3.7 years old, 174.0 ± 8.2 cm, 67.5 ± 8.4 kg) were recruited to participate in this study. All participants received verbal and written information regarding the procedures of the study and provided written consent for their participation in the study. The investigators’ institutional Human Research Ethics Committee approved the study.

### 2.2. Experimental Overview

The study was conducted at Melbourne Arena (Melbourne, Australia), a commonly used arena for team sports competition. Melbourne Arena had previously been fitted with the UWB tracking system and surveyed for calibration of court dimensions. The testing session comprised a team sport movement circuit measuring 15 × 20 m on an indoor parqueted surface. Participants completed the circuit while wearing a receiver tag (ClearSky T6, Catapult Sports, Melbourne, Australia) which was placed in a wearable vest, positioned between the participant’s scapulae. Participants also had attached seven reflective markers, placed on the receiver tag and other prominent landmarks of the participants, and these were used for the reference system to capture participant position and for future reference system analysis. The reference system was set-up around the circuit area, with a larger capture area of 19–24 m to ensure no black spots occurred. All participants completed a self-paced warm-up, before the start of the circuit.

### 2.3. Data Collection

#### 2.3.1. Circuit

The indoor sports movement circuit comprised locomotion activities commonly occurring in indoor sports, as presented in Figure 1. The circuit was designed using research assessing frequently recurring movement sequences to imitate indoor team-sports movements [31]. Participants performed seven movement sequences at self-paced intensities and one maximal acceleration, which was verbally encouraged by a researcher positioned at the beginning of the acceleration station. The movement sequences were performed in the following order:Self-paced walk (9 m).Self-paced jog (9 m).Self-paced jump.Self-paced run (13 m).Maximal acceleration (9 m).Three self-paced 45° changes of direction (13 m).Self-paced side shuffle (15.4 m).Self-paced walk (13m)

#### 2.3.2. Catapult ClearSky T6 Setup

The LPS (Catapult ClearSky T6, Catapult Sports, Melbourne, Australia) previously installed for the area comprised 20 fixed anchor nodes. Nodes were fixed at varying heights ranging between 19.7–20.9 m and proximity from the court boundaries ranging between 12–32 m, as presented in Figure 1; this ensured full court coverage and minimised metal interference. The master anchor was connected via Ethernet cabling to the data processing laptop, which captured data at a reported frequency of 10 Hz. Data were processed using Openfield™ console software version 1.22.2 (Catapult Sports, Melbourne, Australia), with receiver tags worn by participants seen by the system at all times. The system utilises a narrow UWB frequency of range 3.1 to 10.6 GHz to locate receiver tags in the surveyed area. A minimum of three anchor nodes were required to have clear lock on a receiver tag, with the location of the tags being calculated through a multi-process algorithm using two-way ranging (TWR), angle of arrival (AoA) and time difference of arrival (TDOA).

#### 2.3.3. Vicon Setup

The reference system used was a Vicon motion analysis system (©Vicon Motion Systems, Oxford Metrics, UK), set up using 20 cameras (T40 and Vantage) as presented in Figure 1. The system captured at a frequency of 100 Hz, with the cameras mounted on tripods offset 2 m from the perimeter of the circuit area, for a capture area of 19 × 24 m. Seven, 40-mm reflective markers were attached to the receiver tag and other prominent landmarks on the participants:Catapult Unit (receiver tag).Right Shoulder.Left Shoulder.Left Front Hip.Right Front Hip.Right Back Hip.Left Back Hip

The reflective marker attached to the outside of the pouch containing the receiver tag was used as the reference system’s comparative position, while the other six reflective markers were used for future analysis. All 20 cameras were connected via two-gigabit switches that were attached to the data-processing laptop (separate from the LPS laptop) via ethernet cabling. The reference system was calibrated to the capture area, with Vicon calibration image and world errors of 0.094 mm and 0.525 mm respectively. Additionally, the refinement frames were set at 3000 frames with the origin of calibration set using Active Wand v2. Reference system marker dropout was accounted for using Vicon Nexus software version 2.8.2 (©Vicon Motion Systems, Oxford Metrics, UK), by gap filling through automatic pattern detection (maximum 10 frame gaps only filled). This automatically used other marker locations to determine the trajectory of the dropped marker. If these markers were unavailable, the spline fill option was used, which calculates the position based on 10 frames either side of the dropped marker. Finally, when marker dropout was for a substantial length, the data were excluded from the analysis.

### 2.4. Data Processing

Data was exported from the LPS and reference system software and analysed in R statistical software (R: A language and environment for statistical computing, Vienna, Austria). Raw Vicon data was smoothed and filtered using a proprietary Butterworth and moving average filters, to mimic the same processing that is applied to the ClearSky data (further details of smoothing and filtering processes are protected by a non-disclosure agreement). As Vicon data were captured at 100 Hz, compared to Catapult captured at 10 Hz, raw Vicon data were down-sampled from 100 to 10 Hz by sub-setting every 10th frame. Each subset of data was inspected for best fit to the Catapult data. Additionally, the Y component of Vicon data required translation, as it had been captured as the *Z*-axis. Therefore, by finding the mean between the Vicon data and Catapult dataset, Vicon data were translated down to the same scale

### 2.5. Statistical Analysis

Inter-unit distance was calculated for each combination of participants as the distance between each players x, y coordinates. Each participant combination was used once only, resulting in 21 individual combinations (one participant was not used due to poor Vicon data quality). This was calculated for both ClearSky and Vicon datasets using the formula below:(1)$$D=\sqrt{{\left(a_{x}-b_{x}\right)}^{2}+{\left(a_{y}-b_{y}\right)}^{2}}$$
where *D* is the distance between the two participants, *a* is participant one and *b* is participant two and *x* and *y* are the coordinates. The two datasets, ClearSky and Vicon were visually inspected to ensure they lined up at a common starting point (Figure 2).

Criterion validity was measured using root mean square error (RMSE), reported in metres using the following equation:(2)$$RMSE=\sqrt{\frac{\Sigma_{i=1}^{n}{\left(P_{i}-O_{i}\right)}^{2}}{n}}\;\;\;\;\;\;\;\;\;$$
where *P* is ClearSky data; *O* is Vicon data and *n* the length of the time series. Mean bias was used to measure the bias of the ClearSky LPS, and it was calculated using the following formula:(3)$$MB=\frac{1}{n}\overset{n}{\underset{i=1}{\sum }}\;\left(P_{i}-O_{i}\right)$$
where *P* is ClearSky data, *O* is Vicon data and *n* is the length of time series. Finally, the percentage of variance accounted for (%VAF) was used to measure the portion of the variance for Vicon, accounted for by ClearSky. This was calculated for each combination and distance band using the formula below:(4)$$\%\;VAF=100\times \left(1-\frac{\sum_{t=1}^{n}\;{\left(O_{t}-P_{t}\right)}^{2}}{\sum_{t=1}^{n}\;{\left(O_{t}\right)}^{2}}\right)$$
where *P* is Catapult data, *O* is Vicon data, *n* is length of time series and *t* is the time.

Additionally, a rolling RMSE function was used on all combinations, providing a matrix of RMSE as a function of cumulative time. This was used to compute RMSE stabilisation at a threshold of 1/500 of the final rolling cumulative RMSE (Figure 3), to ensure enough data were analysed, whereby the error rates stabilises. Through this stabilisation analysis, data lengths between 43 and 50 s were found to be sufficient for stabilisation of error. All combination datasets were cut at 50 s to ensure consistent results. Finally, analysis was conducted on the association between distance of units and its function on the accuracy of inter-unit distance. Distances were discretised into five interval bands to highlight differences between smaller and larger inter-unit distances on the RMSE, mean bias and %VAF between the LPS and reference system. The accuracy measures were calculated for each distance band, by analysing the values that resided within each band. The five bands were:
0–5 m5–10 m10–15 m15–20 m>20 m

The study methodology was written following a recently published protocol [32] in order to warrant the strict description of the use of technology, scoring 16 points out of 21 (76%). The authors confirm that the data supporting the findings of this study are available within the articles Appendix A.

## 3. Results

The overall RMSE between the Catapult LPS and Vicon system for inter-unit distance was 0.20 ± 0.05 m, while the mean bias was 0.10 ± 0.06 m. Comparisons between ClearSky and Vicon inter-unit distance at different distance bands is presented in Table 1. Inter-unit distance based on distance bands resulted in larger RMSE values at larger distances. Bands of 5–10 m, 10–15 m, 15–20 m and >20 m had RMSE values in the range of 0.20 to 0.22 m, compared to the 0–5 m band with a RMSE of 0.18 m.

## 4. Discussion

The objective of this investigation was to assess the criterion validity of the Catapult ClearSky T6 local positioning system for assessing inter-unit distance. The overall results of the study returned a mean RMSE of 0.20 ± 0.05 m, which was more favorable compared to a previous investigation of a Bluetooth Low Energy Channel tracking system, presenting a mean error of 0.30 ± 0.13 m [7]. The current study also found a mean bias of 0.10 ± 0.06 m, including all distance bands displaying bias overestimates of the true values, especially at distances below 10 m. Finally, %VAF analysis was stable across all distance bands, excluding distances above 20 m. These findings are important for the use of the LPS to accurately measure spatiotemporal variables, as these variables’ base function is centered on inter-unit distances.

These results align with similar research which found the ClearSky T6 LPS to have mean error of 0.21 ± 0.13 m for measuring position [8] in the optimal setup. Previous investigations into the errors associated with anchor placement on validity of the ClearSky system have found increased error in-sub optimal setups (1.79 ± 7.61 m) compared to optimal setups (0.21 ± 0.13 m) for position estimates [8]. This was attributed to node positions, near corners and proximity between node and court boundaries which could reduce accuracy due to increased multipath propagation [33]. Errors of this nature were mitigated as setup was optimized within the stadium, as seen in Figure 1, which represents varying anchor heights to mitigate metal infrastructure interference and adequate proximity of node to edge of the field.

Analysis of the associations between different distance bands and inter-unit distance accuracy indicates increased error at larger distances. These findings suggest that as distances between units increase so does the error of observed values, as seen with a linear increase in RMSE results from 0.18 ± 0.08 at distances between 0–5 m to 0.22 ± 0.05 m above 20 m. It is difficult to compare these results with previous studies, as to our knowledge this study is the first to analyse inter-unit distance accuracy at distances above 20 m. A previous investigation found higher accuracy for larger distances, however the studies distances only ranged from 0.5–1.8 m [7]. The limited amount of research assessing the validity of UWB systems over a spectrum of distances covered in indoor team sports highlights the contribution of the current findings in further understanding the capabilities of the UWB system. While the fixed setup of this study was optimal for the stadium used, each stadium requires correct surveying and optimal positioning of anchors to mitigate black spots and interference. This is especially important, as a mobile version of the system is available, which allows for transportation and manual setup at stadiums. However further research is warranted for this version, to ensure validity and reliability during the manual setup. Additionally, research testing the accuracy of a different number of anchors used in the system would provide further understanding of the system’s capabilities and the minimum number of anchors required for a valid accurate signal. Also, while the number of participants (n = 8) provided an adequate 21 combinations for analysis, additional research into unstructured movements using more participants such as small-sided games or match simulations could be warranted. Finally, further validation of tactical variables is required, as only one study thus far has investigated EPTS for measuring tactical variables [13]. Therefore, future research should look at validation of the inter-stadium reliability of EPTS to allow accurate comparison of tactical variables during matches and training.

## 5. Conclusions

With acceptable inter-unit distance accuracy found in this study, as well as adequate ability to measure distance, speed and position [8,12], the ClearSky LPS can be confidently used to capture spatiotemporal tactical variables which can be used to assess team tactical synchronisation, inter-player interactions, and coordination tendencies. The RMSE for inter-unit distance ranged between 0.18–0.22 m for all distance bands, representing acceptable validity at all distances investigated. This opens up opportunities for increased investigation using spatiotemporal tactical variables in indoor sports.

## Figures and Tables

**Figure 1 sensors-20-03693-f001:**
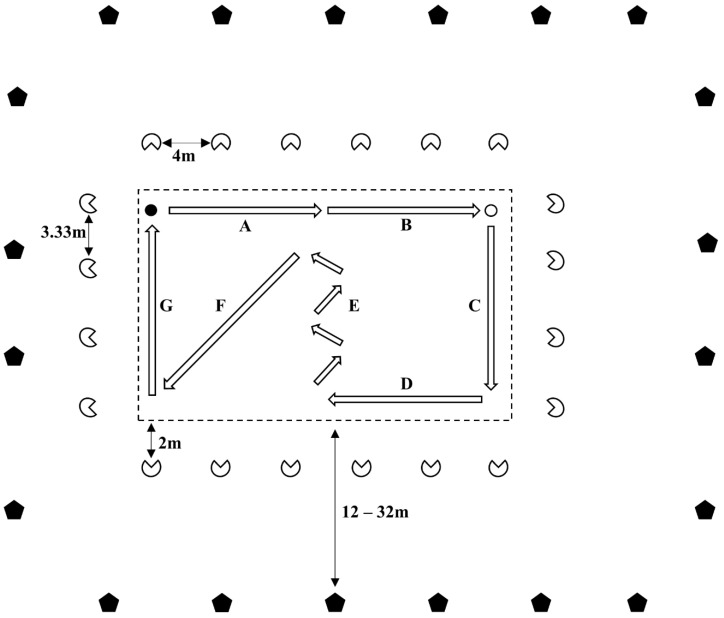
Schematic representation of Vicon setup (white indented circles), ClearSky setup (black pentagons) and circuit boundaries (dashed line), with illustration of circuit movements: start (black circle), walk (**A**,**G**), jog (**B**), jump (white circle), run (**C**), maximum acceleration (**D**), change of direction 45° (**E**) and shuffle (**F**).

**Figure 2 sensors-20-03693-f002:**
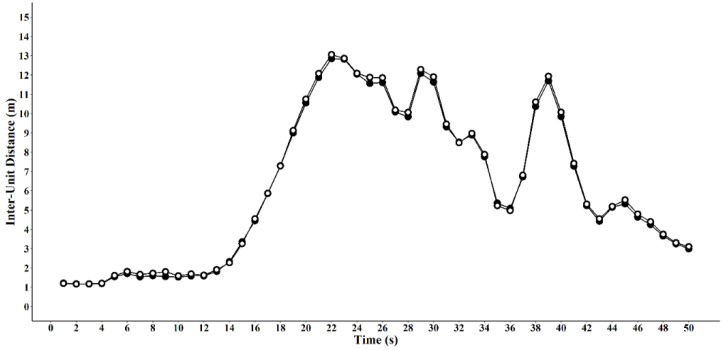
Example comparison of ClearSky (white circles) and Vicon (black circles) inter-unit distance. Vicon data was smoothed and filtered to match Catapult using a proprietary combination of filtering techniques.

**Figure 3 sensors-20-03693-f003:**
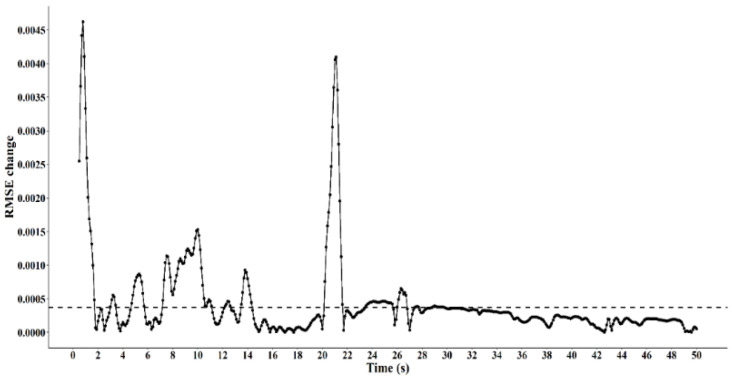
Root mean square error (RMSE) stabilisation. Threshold of 1/500; under threshold change (below dashed line), over threshold change (above dashed line).

**Table 1 sensors-20-03693-t001:** Difference Between Distance Bands’ Inter-Unit Distance Accuracy, Root Mean Square Error, Mean Bias and Percentage of Variance Accounted For.

Distance Band	N. Frames	RMSE (m)	Mean Bias (m)	Percentage of Variance Accounted for (%)
0–5 m	2731	0.18 ± 0.08	0.14 ± 0.10	94.34 ± 0.09
5–10 m	3232	0.20 ± 0.07	0.14 ± 0.10	98.64 ± 0.01
10–15 m	2643	0.20 ± 0.07	0.07 ± 0.06	98.32 ± 0.01
15–20 m	1684	0.21 ± 0.06	0.03 ± 0.05	97.88 ± 0.03
>20 m	210	0.22 ± 0.05	0.06 ± 0.08	74.37 ± 0.28

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
