# Peer review of "Criterion Validity of Catapult ClearSky T6 Local Positioning System for Measuring Inter-Unit Distance"

_sensors, 2020, doi:10.3390/s20133693_

Round 1

Reviewer 1 Report

I only consider that the authors should add to the manuscript some reference regarding the change in the number of anchor nodes and regarding the influence of a portable use of the system in its reliability. That is, the study used 20 anchors. However, it is difficult to use in a competitive environment, for instance when one team plays away, or the system should be fixed in the stadium. Such informations will contribute to clarify the use of such system and its reliability in the future

Reviewer 2 Report

This is a well-written paper that fulfills a gap in the literature, namely a reliable tool to measure interpersonal variables related to tactical team movements in indoor sports. It shows rigorous decisions across the steps of such validation and it transpires sound science.

I have two suggestions to make.

1) tactical variables are not simply spatiotemporal variables. Such measures imply tea behavior concepts to turn spatiotemporal measures into tactics. section 1.3 would benefit from such clarification (see e.g., Araújo, D., & Davids, K. (2016). Team synergies in sport: Theory and measures. Frontiers in Psychology, 7, 1449. doi: 10.3389/fpsyg.2016.01449).

2)In their introduction, the authors rightly indicate that during unstructured movements, high rates of change in velocity and at peak speeds, less favourable results were found concerning the reliability of GPS. In the present study participants performed eight movement sequences such as Self-paced Walk, Self-paced Jog, Jump, Self-paced run, Maximal Acceleration, Three self-paced 45° Change of Directions,  Self-Paced Side Shuffle, Self-Paced Walk. However I couldn't find specific results to the reliability of the tested system in these tasks. These are very important results to present and discuss.

Reviewer 3 Report

Overall is a good article (clean and accurate)

Introduction

Generally, well written. However, I do not understand the importance of section 1.3., namely because it does not provide information about the core concept of the article. I think that it would be more important one section of related work adding information about the validity of LPS systems presented in the literature and create a rationale for the objective of the study. A statement of contribution is missing, and this statement may be developed based a strongly related work. It would be important to highlight limitations or lack of information in previous works and highlight the relevance and pertinence of the present.

Materials and methods

I would like to suggest split section 2.1. in two sections (one of the experimental overview or experimental approach to the problem and other for participants).

Lines 97-98: how did the authors conceived and applied the circuit to ensure that players made the intensity distances as expected?

Section 2.2. add a mention to figure 1 aiming to introduce the circuit. Moreover, classify the “speed intensity” related to walk, jog, and other qualitative info. Additionally, it would be interesting to describe how these “intensities” were ensured.

Results

Possibly it would be interesting to add an additional analysis about the accuracy of the system for each section of the circuit or for different speeds.

Discussion

Generally, the lack of strongly related work was reflected in this section.

Moreover, a section of study limitations and the key-home message is possibly missing.
